# Multimorbidity Patterns and Their Association with Social Determinants, Mental and Physical Health during the COVID-19 Pandemic

**DOI:** 10.3390/ijerph192416839

**Published:** 2022-12-15

**Authors:** Jesús Carretero-Bravo, Begoña Ramos-Fiol, Esther Ortega-Martín, Víctor Suárez-Lledó, Alejandro Salazar, Cristina O’Ferrall-González, María Dueñas, Juan Luis Peralta-Sáez, Juan Luis González-Caballero, Juan Antonio Cordoba-Doña, Carolina Lagares-Franco, José Manuel Martínez-Nieto, José Almenara-Barrios, Javier Álvarez-Gálvez

**Affiliations:** 1Department of Biomedicine, Biotechnology and Public Health, University of Cadiz, Avda. Ana de Viya 52, 11009 Cádiz, Spain; 2Department of Statistics and Operational Research, University of Cadiz, Polígono Río San Pedro, 11510 Puerto Real, Spain; 3Department of Nursing and Physiotherapy, University of Cadiz, Avda. Ana de Viya 52, 11009 Cádiz, Spain; 4Preventive Medicine Area, Hospital of Jerez, Ctra. Trebujena, s/n, 11407 Jerez de la Frontera, Spain

**Keywords:** multimorbidity patterns, COVID-19, mental health, social determinants, physical health, latent class analysis

## Abstract

Background: The challenge posed by multimorbidity makes it necessary to look at new forms of prevention, a fact that has become heightened in the context of the pandemic. We designed a questionnaire to detect multimorbidity patterns in people over 50 and to associate these patterns with mental and physical health, COVID-19, and possible social inequalities. Methods: This was an observational study conducted through a telephone interview. The sample size was 1592 individuals with multimorbidity. We use Latent Class Analysis to detect patterns and SF-12 scale to measure mental and physical quality-of-life health. We introduced the two dimensions of health and other social determinants in a multinomial regression model. Results: We obtained a model with five patterns (entropy = 0.727): ‘Relative Healthy’, ‘Cardiometabolic’, ‘Musculoskeletal’, ‘Musculoskeletal and Mental’, and ‘Complex Multimorbidity’. We found some differences in mental and physical health among patterns and COVID-19 diagnoses, and some social determinants were significant in the multinomial regression. Conclusions: We identified that prevention requires the location of certain inequalities associated with the multimorbidity patterns and how physical and mental health have been affected not only by the patterns but also by COVID-19. These findings may be critical in future interventions by health services and governments.

## 1. Introduction

The emergence in late 2019 of the COVID-19 disease caused an unprecedented health crisis, becoming a leading cause of morbidity and mortality worldwide [1]. Within months, the COVID-19 pandemic spread throughout the world, also generating a significant social and economic crisis [2,3]. The symptomatology of this new disease is very diverse and ranges from asymptomatic infection [4] to severe respiratory tract symptoms and severe pneumonia with multi-organ dysfunction [5]. In addition to all the physical problems associated with this disease, patients with COVID-19 and those populations most exposed to the disease are at an increased risk of psychiatric illness, including cognitive impairment or disorders like anxiety, depression, or insomnia [6,7,8]. In addition, pre-existing mental disorders were associated with severe COVID-19 and increased mortality [9].

Multimorbidity can be defined generically as the accumulation of two or more chronic health conditions [10]. The increasing prevalence of chronic diseases and the rapid ageing of the population has led to an increase in multimorbidity worldwide [11,12], making it one of the greatest challenges for healthcare systems [13,14]. This condition is associated with reduced quality-of-life, increased disability, functional impairment, increased healthcare utilization and fragmentation of care, complex treatment, and increased mortality [15,16,17], with the consequent increase in aspects such as health and social costs and poly-medication. Additionally, there are strong associations between having multimorbidity and poorer mental and physical health outcomes [18,19] and a direct association between having multiple physical chronic diseases and consequent common mental disorders [12,20,21].

In this context, some studies highlighted the incapacity of current clinical guidelines to tackle the complex needs of patients with multimorbidity because of inadequate or modest attention to co-occurring diseases [22,23]. Given this fact, it is necessary to apply new ways of addressing this complex problem. In the last 15 years, numerous studies have emerged intending to detect specific patterns of chronic conditions to deal with the problem of multimorbidity by considering something more specific than just the accumulation of chronic conditions [24,25]. Among pattern detection techniques, several studies have used statistical methods such as Cluster Techniques [26,27] or latent variable techniques such as Factor Analysis [28] or Latent Class Analysis (LCA) [16,29]. A systematic review conducted in 2019 [25] showed the heterogeneous use of these techniques when analyzing chronic diseases. The same study also highlighted the increasing use of the LCA technique in recent years due to its suitability for the biodata associated with chronic diseases and the causality in the relationship between diseases in the same patterns [25].

Additionally, because of the pandemic, multimorbidity has been associated with severe COVID-19 disease resulting in hospitalization, ICU admission, intubation, or death [30,31]. In an analysis of nearly 300,000 confirmed COVID-19 cases reported in the United States, the mortality rate was 12 times higher among patients with reported comorbidities compared to those with none [32]. In an Italian study with 1700 patients, multiple comorbidities were associated with an exponential growth mortality rate, regardless of age [33]. Another Italian study showed that specific comorbidities, such as cardiometabolic comorbidities, worsen the prognosis in COVID-19 patients compared to those with only one cardiometabolic disease [34]. In addition, COVID-19 itself has led to worsening physical and mental health not only for those who have suffered the disease but also for those who have suffered the consequences of the restrictive measures. In fact, these measures have led to a dramatic change in lifestyles and social behaviors of multimorbid people [35,36]. Other studies have shown how people with previous multimorbidity had poorer mental health and less resilience during the pandemic [37,38].

It is known that non-communicable diseases are influenced by social determinants between populations (socio-economic status, educational level, or economic hardship) and lead to a deterioration in both physical and mental health [39,40,41]. Several studies have studied the relationship between social determinants of health and multimorbidity [11,42] and how the most vulnerable groups (low socio-economic status, ethnic minorities, or older people) may be more prone to suffer multimorbidity. In addition, a few other studies have measured these determinants among some multimorbidity patterns [16,28,29], although this relationship must be studied during the pandemic.

With the aim of analyzing multimorbidity patterns in a population affected by the pandemic (the province of Cadiz, in South Spain) and their relationship with physical and mental health, we designed a questionnaire based on the European Health Survey in Spain. We used this survey on people over 50 years of age with three fundamental objectives: (1) to identify multimorbidity patterns among the Cadiz population, (2) to analyze the association between these patterns and variables associated with the pandemic, and (3) to study the association of these patterns with mental and physical health, and possible health determinants.

## 2. Materials and Methods

### 2.1. Design and Setting

This is a cross-sectional and prevalence study based on a self-reported questionnaire carried out in February 2022. The population was people over 50 years in the province of Cádiz, Southern Spain. We conducted the survey through telephone calls, so the accessible population included only people with a telephone available. It has been studied that in terms of health issues, telephone interviews produce comparable results to face-to-face interviews while allowing access to a larger number of subjects [43,44].

The province of Cádiz is a region in Southern Spain with a population of around 1,250,000 inhabitants that live mainly in municipalities with more than 30,000 inhabitants; around 15% of the province lives in rural areas [45]. Being part of Spain, it has a universal and free health care system, but it is one of the provinces with the worst unemployment rates and most remarkable socio-economic inequalities in Spain [46]. As in the rest of Spain, the incidence of the COVID-19 pandemic in Cádiz has been considerable, causing the collapse of health services and an increase in expected mortality.

We carried out a stratified sampling, considering the six administrative regions of the province. We conducted telephone interviews until we reached the quotas per region. We estimated that for a confidence level of 95% and an estimation error of ±2.5 units, a sample of 1600 people would be necessary. This sample was extended by 600 extra people living in the biggest city in the province, Jerez de la Frontera, to carry out an in-depth analysis of this city. The response rate for the telephone survey was 13.3%. From the initial sample, we selected those with two or more chronic conditions from the 33 conditions we asked for, as other studies of multimorbidity patterns have done [24,25]. After this selection, the final sample for analysis consisted of 1592 individuals, 72.4% of the sample.

This questionnaire is part of the DEMMOCAD (Determinants of Multimorbidity in Cadiz Province) project, which aims to conduct an in-depth analysis of multimorbidity profiles in the province of Cadiz. The relevant ethics committee has accepted the DEMMOCAD project for collecting the survey data.

### 2.2. Measures

The self-developed questionnaire was mainly based on items from the European Health Survey conducted in Spain in 2020 [47]. This survey is a 5-yearly study that analyses the health status and associated social determinants in the Spanish population.

In our questionnaire, we obtained three sections of interest for our research. We asked for 32 chronic conditions listed in the European Health Survey (see Appendix A), in addition to the weight and height to calculate whether the individual was obese.

We used the items of the SF-12 (Short Form Health Survey) scale to analyze the physical and mental health of the population [48]. It is a shortened version of the SF-36, a scale that has been widely validated in various populations and consists of eight dimensions that measure various aspects of quality-of-life related to physical and mental health. The SF-12 version is usually associated with two dimensions (Mental Health and Physical Health) and has an algorithm that allows scoring health in these two dimensions on a scale from 0 to 100 [49].

We also asked about variables associated with the pandemic situation, such as COVID-19 diagnosis, possible after-effects of having COVID-19, and the use of health services in the last year (visits to primary care and emergency services, possible visits to a specialist doctor, and hospital admissions).

Finally, we obtained variables about socio-economic determinants, such as the administrative region and the city in Cádiz province, gender, age (50–59, 60–69, and older than 69), disability, education (none, primary, secondary, and university), income level (less than 600 €, 601–900 €, 901–1200 €, 1201–1800 €, and bigger than 1800 €), and employment status (active, retiree, unemployed, and domestic work). We also obtained lifestyle variables, such as fruit and vegetable consumption (one per week or less, four times per week, one per day, two or more per day), alcohol consumption (never, one per month or less, one per week or less, several times per week), tobacco consumption (neither smokes nor has smoked, used to smoke, 1–10 cigarettes, and >10 cigarettes), and physical activity (never, once per month or less, several times per week, all days).

### 2.3. Data Analysis

Firstly, a descriptive analysis of the sample was carried out (means and SDs for continuous variables, frequencies and percentages for categorical). We analyzed the missing values and the categories associated with non-response for each variable. We found missing values for some variables: education, disability, job situation, and income level. For the first three variables, the missing values were less than 1% of the sample size, and we imputed it. Only in the income level variable we considered the category ‘do not know/no answer’ in the model with covariates due to the high number of responses in this category (12%).

We used Latent Class Analysis (LCA) to identify multimorbidity subgroups with interdependent disease patterns. LCA is a multivariate technique used to classify observations based on patterns of categorical responses. While there are other techniques that may be appropriate in this context, such as some clustering techniques [26,27], LCA is a well-suited technique for our data, given the binary variables associated with chronic conditions and the causal relationship we hypothesized between the conditions in each pattern. We created the statistical model in LCA with those conditions with more than 2% prevalence. Therefore, from the initial 33 conditions, we discarded ‘other mental problems’ and ‘cirrhosis, liver dysfunction’, so we performed LCA on 31 chronic conditions.

We obtained the LCA model following four steps. In the first selection stage, we established the number of classes appropriate, considering four criteria [50]. Firstly, we reviewed model goodness-of-fit indices, considering BIC, ABIC, CAIC, and entropy, taking ABIC as a reference due to it tends to detect the number of classes better in not very large sample sizes [51]. The lower the indices, the better the model and the higher the entropy, the better the variance is explained. We also checked the *p*-value of the bootstrap likelihood ratio test (BLRT), a test to check the fit of the model in an absolute way [52]. As a second criterion, we selected classes with enough participants (n > 50 and 5% of the sample size [50]). In addition, we reviewed the probability of belonging to each class and the clinical interpretability, with this non-statistical criterion being equally important to the others.

In the second phase, we measured mental and physical health with the SF-12 scale. At first, we analyzed their psychometric properties. For this purpose, we calculated the internal reliability through the global Cronbach’s Alpha coefficient and the partial Alpha coefficient, obtained by eliminating each item individually, with acceptable values above 0.7 [53]. We also analyzed the construct validity of the SF-12 scale in the population over 50 years by means of an Exploratory Factor Analysis (EFA) and a Confirmatory Factor Analysis (CFA). We checked the adequacy of the data to EFA through the KMO statistic and Bartlett’s test of sphericity [54]. We transferred the dimensions obtained in the EFA to a CFA, where we confirmed the validity of the structure with the Comparative Fit Index (CFI) and the Root Mean Square Error of Approximation (RMSEA). The values of these indices range from 0 to 1, and CFI > 0.9 and RMSEA < 0.5 were acceptable [55].

Next, we obtained the score of the two dimensions of the SF-12 instrument and measured the differences among patterns using the ANOVA test and the Tukey HSD test for a posteriori differences. In addition, we analyzed the variables associated with COVID-19 within each pattern and made a comparative analysis of the mental and physical health of each pattern according to the variables associated with the pandemic. Finally, we introduce the SF-12 scores, the variables associated with COVID-19, and the other socio-demographic covariates of interest in a multinomial logistic regression model using the three-step approach, a recommended technique to minimize class shifts and adjust for potential bias due to the probabilistic nature of latent class [56].

We performed LCA using Mplus 7.2. We conducted other analyses and tables using RStudio and R. We generated some graphics with the ggstatsplot package [57].

## 3. Results

The descriptive statistics by COVID-19 diagnosis can be found in Appendix A. There were 1592 individuals with multimorbidity with a mean of 4.77 conditions. Most of the individuals were women, who accounted for 57%, and the distribution by age group was similar, with all three age groups accounting for more than 30%. Regarding COVID-19, as of February 2022, the end of data collection, 19% of the sample had a positive diagnosis.

### 3.1. Model Selection of Multimorbidity Patterns

We selected the number of latent classes at the individual level considering the BIC, ABIC, CAIC, entropy, and BLRT test for the 2–8 class models (Table 1). BLRT values show an adequate fit in all models. The lowest ABIC value occurred in the five-class model, while BIC and CAIC were better in the three-class model. Since ABIC is more sensitive to small sample sizes, the five-class model was analyzed.

This model fulfilled the other three requirements. On the one hand, the patterns obtained were clinically consistent and sufficiently different to respect the parsimony. Regarding the sample size, each class was around 5% and bigger than 50, with the smallest class being 76 people. Finally, we found that 90% of the individuals had a maximum probability of more than 58% to stay in one class. Thus, we chose the five classes model, with an entropy value of 0.728.

Considering the overall prevalence of conditions, as shown in Figure 1, and the prevalence of each condition in Appendix A, the first class is people ‘Relatively Healthy’ (n = 448), without a clear pattern of conditions and with few conditions, the second class is associated with ‘Cardiometabolic’ conditions (diabetes, hypertension, obesity, and cholesterol, n = 659), the third class is people with ‘Musculoskeletal’ conditions (chronic back pain both lumbar and cervical and osteoarthritis, n = 247), the fourth class is a joint pattern between ‘Musculoskeletal and Mental’ conditions (arthrosis, cervical and lumbar pain, depression and anxiety, n = 76) and the last class is a ‘Complex Multimorbidity’ pattern (n = 162), with cardiometabolic and musculoskeletal conditions, allergies, cataracts, or arthrosis.

### 3.2. SF-12 Scale Properties and Relations

We first calculated the internal reliability of the scale. We obtained a Cronbach’s Alpha of 0.86 for the overall scale, and the Alpha values if we removed one item ranged from 0.83 to 0.85, all of which were above the 0.7 fit cutoff. Appendix A shows the factor structure obtained in the EFA. The KMO statistic (0.88 > 0.7) and Bartlett’s test of sphericity (χ^2^ = 14,154.11, *p*-value < 0.000) showed good suitability for such an analysis, and the proposed factor structure is comparable to the original instrument.

We transferred the structure obtained in the EFA to a CFA, where we incorporated the necessary inter-item covariances, and we calculated the fit indices. The final model can be seen in Figure 2. We obtained an RMSEA = 0.047, adequate to the condition of being less than 0.05 and a CFI = 0.985, which met the criterion of being greater than 0.90, so we can speak of a scale with adequate construct validity to measure mental and physical health in this population. Within the MCS component, the item with the bigger weight is MH2 (referring to discouragement and sadness during the last few weeks), while in the physical component, the item with the bigger weight is BP (referring to bodily pain during the last week). We also observed a significant correlation between the two dimensions.

Following this analysis, we obtained the values of the mental dimension (MCS) and the physical dimension (PCS) of the SF-12 scale on a scale from 0 to 100 using the procedure described in [49]. We compared them among the five multimorbidity patterns (Figure 2). In the MCS dimension, the mean values ranged from 38.28 (in the ‘Musculoskeletal and Mental’ pattern) to 44.21 (in the ‘Cardiometabolic’ pattern), while in the PCS they ranged from 35.32 (in the ‘Complex Multimorbidity’ pattern) to 40.49 (in the ‘Relative Healthy’ pattern). The ANOVA test showed significant differences between patterns in both the MCS component (*p*-value < 0.000) and the PCS (*p*-value < 0.000). The a posteriori differences in Tukey’s test showed that the patterns with significantly lower mental and physical health were ‘Complex Multimorbidity’ and ‘Musculoskeletal and Mental’, which also accumulate the most conditions (Figure 3).

Additionally, we compared the values of the MCS and PCS dimensions within each pattern according to whether they had a positive COVID-19 diagnosis using the Student’s t-test (Table 2). We observed a significant decrease in the ‘Relative Healthy’ class in both the mental and physical dimensions, while in the ‘Cardiometabolic’ class there was a significant decrease in the MCS component. Patterns with more conditions showed no significant changes in both health dimensions. Finally, regarding the incidence of COVID-19 and COVID-19 sequelae, there were no significant differences between patterns (Appendix A), although the ‘Musculoskeletal and Mental’ class showed a higher prevalence of COVID-19 after-effects (although not significative).

### 3.3. Social Determinants and Multimorbidity Patterns

Appendix A summarizes the social determinants of the sample individuals in each class and the distribution of the clusters in each pattern. Table 3 shows the multinomial logistic regression results associating the covariates with the class of membership in each condition, with the ‘Relative Healthy’ pattern taken as a reference, as it is the one with the fewest conditions (3.47) and does not have a clear specific pattern. All the patterns, except the ‘Cardiometabolic’, show a significant decrease in health in both dimensions with respect to the ‘Relative Healthy’ pattern.

By gender, men have significantly more risk of being in the ‘Cardiometabolic’ pattern, while by age, ‘Cardiometabolic’ and ‘Complex Multimorbidity’ patterns have older members, especially in the last group (OR of 7.924 and *p*-value < 0.001 in the group aged over 69 years). We can also see a significantly higher percentage of disabled people in ‘Complex Multimorbidity’ and ‘Musculoskeletal’ groups.

If we talk about socio-economic determinants, we see differences in the level of education among the patterns, with the ‘Complex Multimorbidity’ pattern having a lower level of education. Additionally, the ‘Musculoskeletal and Mental’ pattern had a significantly lower income level than the other patterns. We did not find significant differences in the patterns according to the region of the province of Cadiz.

Variables associated with individuals’ lifestyles were also analyzed. While the consumption of fruit and vegetables was not significantly associated with any pattern, we found associations with physical activity, where an increase is a factor related to less prevalence in the ‘Musculoskeletal and Mental’ pattern, and with alcohol consumption, with significantly lower consumption also in the ‘Musculoskeletal and Mental’ pattern. Tobacco consumption was also a risk factor in the ‘Musculoskeletal’ group.

Finally, regarding health services, we observed that the ‘Complex Multimorbidity’ pattern makes more use of primary care, emergencies, and hospital admissions, while the ‘Musculoskeletal’ pattern makes significantly more use of primary care. In contrast, the ‘Cardiometabolic’ pattern makes less use of specialist services.

## 4. Discussion

This study aimed to identify multimorbidity patterns among the Cadiz population and to analyze the association between these patterns, mental and physical health, variables associated with the COVID-19 pandemic, and health determinants. To this end, we have applied the Latent Class Analysis (LCA) model to 31 chronic conditions and obtained five different patterns among people with multiple chronic conditions.

It is essential to highlight the use of LCA as a pattern detection technique. In a review in 2018 [58], LCA is listed as an underused technique for pattern detection, while in another 2019 review [25], although the most common were Cluster techniques or Factorial Analysis, there was an increase in the use of LCA, making it a growing technique for locating groups of chronic diseases in patients with multimorbidity. Compared to previous techniques, LCA provides a probabilistic approach, which makes it possible to associate in everyone a probability of belonging to each multimorbidity pattern. Moreover, it is a technique with good statistical theoretical development. While we made the choice of the model that best fits our data through the goodness-of-fit criteria in our results, we could discuss the overall quality of our model.

The entropy is better closer to 1. Although it does not reach the usual criterion of being higher than 0.8, it has a value of 0.728, which is higher than the minimum acceptable value of 0.6 [50] and comparable to other studies of LCA in multimorbidity patterns [59,60]. We have also analyzed the average latent class posterior probabilities. The acceptable criterion is that the values of the diagonal of the matrix with the probabilities must be bigger than 0.8 [50]. In our case, the values oscillate between 0.794 and 0.843, with only one class below 0.8 (but very close); therefore, we obtained adequate values.

Regarding the prevalence of chronic conditions, it is true that 72.4% of people aged over 50 years had more than two conditions, which is higher than other studies in Spain [61] and that proposed globally [62]. However, we must consider that this prevalence highly depends on the number of conditions (in our case, 33 conditions) and the type of data collection [25]. Concerning the European Health Survey in Spain, on which our study is based, the results appear similar (64.7% with two or more chronic conditions).

When analyzing the patterns obtained, they were comparable with other studies. In a review in developed countries [58], the three most common patterns were cardiovascular, mental health, and allergic disease, with two of these three patterns being partially present in our study. In another review of 51 studies [25], cardiometabolic and mental health patterns were detected as the most frequent. In addition, in the same review, patterns detected specifically through LCA were analyzed, and a musculoskeletal pattern was frequently found. Therefore, three patterns in our research are among the most widespread in other studies.

Although it does not appear among the most frequent in previous reviews, the ‘Complex Multimorbidity’ pattern appears in other studies and is associated with the progressive ageing of society [29,63]. It has the most chronic conditions on average (9.80) and groups parts of several other patterns such as musculoskeletal, cardiovascular, allergies, and mental conditions. It is a fundamental pattern in understanding the challenge that multimorbidity poses in terms of expenditure for health systems in Spain.

The two patterns with people with fewer conditions (‘Relative Healthy’ and ‘Cardiometabolic’) represent around 69% of the population in our study, which is consistent with other studies showing that patterns with fewer conditions represent 63.8% [29]. The ‘Cardiometabolic’ pattern, made up of conditions associated with metabolic syndrome such as hypertension, high cholesterol, obesity, and diabetes, appears in several studies in Spain and is also prevalent in our ‘Complex Multimorbidity’ pattern. The patterns with the most conditions and the highest healthcare represent 31% of our sample and indicate the challenge of multimorbidity in people aged over 50 years.

One of the main contributions of our research is to relate physical and mental health with the groups of conditions obtained in the context of the pandemic. Firstly, it is important to analyze the properties of the instrument used, the SF-12. This is a widespread instrument measuring quality-of-life health in individuals with multimorbidity [64,65], widely validated in Spain [66,67]. Our study shows a factorial structure comparable to the SF-12 original scale [49] and similar to other studies in our country [66].

If we talk about the relationship of health status with multimorbidity, we observed a clear trend: the higher the number of conditions in the patterns, the worse the values on the SF-12 scale in its two dimensions, a trend that is confirmed in other studies with health quality-of-life [16,68,69,70]. Significant is the difference in our study in the patterns ‘Musculoskeletal and Mental’ and ‘Complex Multimorbidity’, with the lowest scores in both dimensions. A study in Denmark with multimorbidity patterns showed that ‘Complex Metabolic’ and ‘Complex Respiratory’ patterns were significantly associated with poorer scores in both dimensions of the SF-12 [16], while another study in Japan found that cardiovascular and digestive patterns were associated with a reduction in the physical component of the SF-36. Our findings, together with those of the other studies, show how specific patterns, which may be influenced by social determinants, further increase the differences in individuals’ self-perceived health status.

In this pandemic context, our study also categorizes possible worsening of the SF-12 instrument according to the COVID-19 diagnoses. In the ‘Relative Healthy’ class, people with COVID-19 had lower scores on both dimensions of health, while in the ‘Cardiometabolic’ class, the reduction in health was significant only in the mental dimension. These are the patterns with fewer conditions, even though they are not people in optimal health. The pandemic appears to have reduced the physical and mental health of people with multimorbidity [35,37], and our study shows how the situation derived from COVID-19, which has led to a social and economic crisis [2,3], may have decreased self-reported health in those individuals who have recovered from COVID-19 and had fewer chronic conditions (and less age), as the pandemic has affected the health quality-of-life (primarily mental) of younger people, with more social relations and of working age [71,72].

A final issue of interest associated with COVID-19 is the after-effects that it will have, especially given that little is known about it so far. Although there have been no significant results, the ‘Musculoskeletal and Mental’ pattern has shown a higher percentage of individuals with sequelae of COVID-19. Other studies have shown how specific conditions can lead to more sequelae of COVID-19 [73], but more studies are needed that focus mainly on the issue of multimorbidity patterns and this virus.

In addition to the problems generated by the pandemic, the multimorbidity patterns in our study show relationships with specific social determinants. For example, the ‘Cardiometabolic’ pattern is associated with older people, mainly men, who have the highest prevalence of metabolic syndrome in Spain [74].

We found an association between tobacco consumption and the ‘Musculoskeletal’ pattern. Tobacco is one of the leading causes of health problems worldwide, and other studies have also shown this relationship in both genders [59] and in middle-aged women [75]. The ‘Musculoskeletal and Mental’ pattern is significantly associated with lower physical activity and income. Patterns with a mental component tend to be associated in other studies with lower purchasing power, lower educational level, and poorer quality-of-life [76,77], with mental patterns being among the most frequent in people of all ages [25,58]. We can explain the lower physical activity found in the ‘Musculoskeletal and Mental’ pattern by the limitations derived from physical pathologies, as well as by the low motivation, apathy, or avoidance behaviors typical of depressive or anxious symptoms. They also consume less alcohol, which may be influenced by medications associated with depression and anxiety. In this sense, it would be desirable for professionals in contact with these people to integrate social prescription as a counterpoint to the use of these drugs [78], and it is also a valuable means of normalizing social relations now that the vaccine has made it possible to recover previously lost habits.

Finally, the ‘Complex Multimorbidity’ pattern is associated with older and less educated people compared to the other patterns. It is a pattern related to older age in other studies, in addition to low-income levels [29] and low education [61,79], thus higher education was significantly associated with a decreased risk of general multimorbidity, being the pattern with the highest number of chronic conditions. Given the problem of ageing in Spain, with several regions having the highest life expectancy rates in Europe, finding the characteristics that lead to this type of pattern is essential to address the challenge of multimorbidity.

Finally, our study also looked at the use of health services. The three patterns with more conditions have more visits to both primary and emergency care, with the difference in the ‘Complex Multimorbidity’ and ‘Musculoskeletal’ patterns being significant. The increase in health services is one of the consequences of these patterns, as shown in other studies [80,81], and is one of the keys for the excessive healthcare expenditure that the problem of multimorbidity entails. Moreover, in the case of complex patterns, this excessive use of health services is associated with problems of poly-medication and diagnoses [22,82]. In the case of the ‘Cardiometabolic’ pattern, the use of specialist services is lower, probably due to this pattern having chronic conditions usually treated by primary care services. Primary care also takes on most musculoskeletal pathologies in the first instance, usually associated with mild pain or functional limitation, before referral to specialists, and probably for this reason these patterns have significantly higher use of these services.

### Strengths and Limitations

The main strength of this study is that it is one of the first studies, to our knowledge, to analyze multimorbidity patterns and their relationship with quality-of-life in health and social determinants in the context of the pandemic in Spain. In the absence of research, this study provides essential information on the distribution of multimorbidity patterns in an area of particular importance, one of the most socially unequal areas of the Iberian Peninsula, hard hit by the pandemic. The questionnaire used for the study is based on a widely applied survey in Spain and covers the most important health characteristics in the most summarized form possible.

However, this study also has certain limitations. Firstly, we obtained conditions in a self-reported way, which can lead to biases, as the prevalence of each chronic condition may be underestimated. As this is a cross-sectional study, the reported results do not show causality and asking about the chronic conditions all at once does not allow for a longitudinal analysis regarding the appearance of the conditions in each multimorbidity pattern. Moreover, as this was a cross-sectional study, changes in the physical and mental health of people with multimorbidity before and after the pandemic cannot be analyzed. In addition, while the sample size is large enough for certain findings, it would be interesting to increase it to reach a larger sample of people who have recovered from COVID-19. In future studies, a larger sample size per district would serve to strengthen our initial results, enabling us to make associations with areas within our province, which in this study have not shown significant differences. In addition, a larger sample size will allow us to detect more individuals with possible COVID-19 sequelae and to know the relationship of these sequelae with previous multimorbidity patterns.

## 5. Conclusions

One of the fundamental tasks in the current challenge posed by multimorbidity is to obtain elements that allow for interventions based on prevention and associated with the most disadvantaged groups. To this end, it is necessary to detect the pattern of association of the different chronic conditions (physical and mental) which generate multimorbidity and their relationship with mental and physical health in this situation caused by the COVID-19 pandemic.

The findings of our study are the first basis for outlining the patterns of multimorbidity in our province and how these have been affected by the pandemic. By locating certain social inequalities associated with the patterns and how physical and mental health has been affected not only by the patterns but also by COVID-19, we are able to identify prevention needs. In fact, these social determinants that show a relationship with patterns (gender, low income, educational level, or lifestyles) may be critical in future interventions by health services and governments, with the fundamental aim of preventing the onset of multimorbidity with age, increasing the efficiency of the health system, and reducing the costs that chronic diseases may entail.

## Figures and Tables

**Figure 1 ijerph-19-16839-f001:**
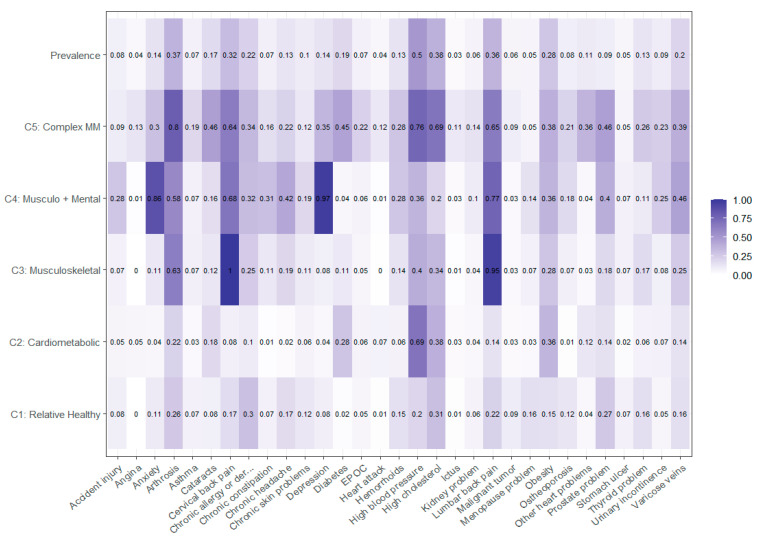
Final model of chronic conditions distribution in five classes and overall prevalence.

**Figure 2 ijerph-19-16839-f002:**
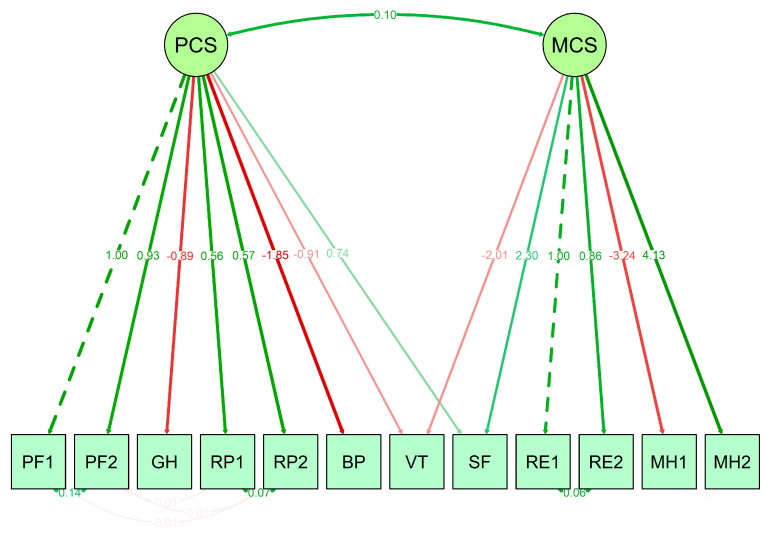
Structure of the SF-12 Scale in the CFA. The values show the coefficients of the model. We name the items according to the dimension of SF-36, PF = Physical Functioning, GH = General Health, RP = Role Physical, BP = Bodily Pain, VT = Vitality, SF = Social Functioning, RE = Role Emotional, MH = Mental Health.

**Figure 3 ijerph-19-16839-f003:**
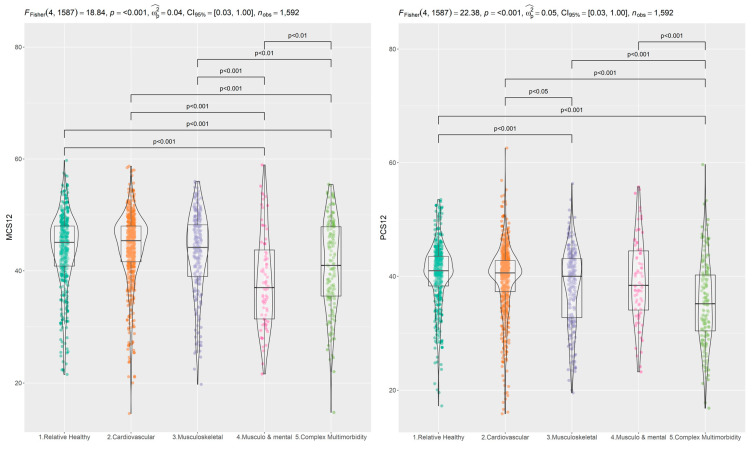
Boxplot of SF-12 dimensions among patterns and ANOVA test and a posteriori comparison in each dimension. MCS: Mental Component Score, PCS: Physical Component Score.

**Table 1 ijerph-19-16839-t001:** Goodness of Fit Values for Models with 2–8 Classes.

Classes	CAIC	BIC	ABIC	Entropy	BLRT *p*-Value
2	36,327.68	36,264.68	36,064.54	0.763	0.44
3	36,250.13	36,155.13	35,853.33	0.651	0.46
4	36,321.47	36,194.47	35,791.02	0.703	0.40
**5**	**36,436.36**	**36,277.36**	**35,772.25**	**0.728**	**0.32**
6	36,579.01	36,388.01	35,781.24	0.700	0.40
7	36,744.60	36,521.60	35,813.17	0.720	0.44
8	36,920.61	36,665.61	35,855.53	0.739	0.22

The selected model is marked in bold.

**Table 2 ijerph-19-16839-t002:** Mental and Physical Health among Patterns depending on COVID-19.

	C1—Relative Healthy	C2—Cardio-Metabolic	C3—Musculo-Skeletal	C4—Musculo-Skeletal and Mental	C5—Complex Multimorbidity
MCS ^1^					
NO COVID-19	44.171	44.526	43.358	38.238	41.017
COVID-19	42.253	42.918	42.530	38.450	40.734
	t = −2.274 *p* = 0.024 *	t = −2.391 *p* = 0.017 *	t = −0.732 *p* = 0.466	t = 0.100 *p* = 0.921	t = −0.149 *p* = 0.883
PCS ^2^					
NO COVID-19	40.806	39.618	37.694	39.094	35.466
COVID-19	39.025	39.369	38.357	39.350	34.699
	t = −2.506 *p* = 0.013 *	t= −0.424 *p* = 0.672	t = −0.613 *p* = 0.541	t = −0.123 *p* = 0.901	t = −0.466 *p* = 0.643

^1^ Mental Component Score; ^2^ Physical Component Score; * Significative difference.

**Table 3 ijerph-19-16839-t003:** Multinomial Logistic Regression of social determinants associated with patterns of chronic conditions. OR referred to Relative Healthy class.

C1—Relative Healthy (Ref.)	C2—Cardio- metabolic	C3—Musculo- skeletal	C4—Musculo- skeletal and Mental	C5—Complex Multimorbidity
Category	OR (95% IC)	OR (95% IC)	OR (95% IC)	OR (95% IC)
MCS12 (mental)	0.98 (0.956, 1.005)	0.949 (0.921, 0.977) ***	0.949 (0.907, 0.992) *	0.892 (0.861, 0.924) ***
PCS12 (physical)	0.986 (0.964, 1.009)	0.971 (0.948, 0.996) *	0.908 (0.874, 0.942) ***	0.911 (0.882, 0.941) ***
**Gender**				
Female (Ref.)	1	1	1	1
Male	3.013 (2.193, 4.141) ***	1.197 (0.793, 1.807)	1.233 (0.611, 2.488)	1.134 (0.644, 1.996)
**Age**				
50–59 (Ref.)	1	1	1	1
60–69	2.06 (1.439, 2.95) ***	1.465 (0.963, 2.228)	0.78 (0.385, 1.581)	2.871 (1.474, 5.593) **
>69	3.417 (2.184, 5.346) ***	1.597 (0.922, 2.765)	1.001 (0.419, 2.392)	8.159 (3.875, 17.183) ***
**Disability**				
Yes (Ref.)	1	1	1	1
No	0.943 (0.63, 1.413)	2.086 (1.329, 3.275) **	1.016 (0.487, 2.118)	1.967 (1.147, 3.371) *
**Education**				
No Education (Ref.)	1	1	1	1
Primary	0.909 (0.571, 1.447)	1.317 (0.723, 2.398)	0.985 (0.424, 2.288)	0.811 (0.447, 1.473)
Secondary	0.843 (0.51, 1.392)	1.047 (0.549, 2)	1.089 (0.432, 2.742)	0.358 (0.171, 0.746) **
University	0.831 (0.472, 1.462)	0.827 (0.386, 1.77)	1.332 (0.437, 4.063)	0.318 (0.128, 0.791) *
**Fruits and Vegetables**				
One per week or less (Ref.)	1	1	1	1
4 times per week	0.965 (0.583, 1.596)	1.073 (0.57, 2.02)	1.461 (0.533, 4.004)	0.985 (0.447, 2.171)
One per day	0.891 (0.534, 1.487)	1.174 (0.618, 2.233)	0.803 (0.265, 2.439)	0.859 (0.378, 1.953)
Two or more per day	0.91 (0.552, 1.5)	0.986 (0.525, 1.853)	1.325 (0.479, 3.661)	0.896 (0.403, 1.991)
**Physical Activity**				
Never (Ref.)	1	1	1	1
One per month or less	1.127 (0.623, 2.039)	1.215 (0.606, 2.436)	0.773 (0.278, 2.147)	0.794 (0.29, 2.174)
Several times per week	0.912 (0.644, 1.292)	0.804 (0.525, 1.232)	0.353 (0.171, 0.729) **	0.768 (0.446, 1.323)
All days	1.064 (0.747, 1.514)	0.773 (0.496, 1.203)	0.395 (0.194, 0.804) **	0.849 (0.489, 1.472)
**Alcohol Consumption**				
Never (Ref.)	1	1	1	1
One per month or less	1.251 (0.711, 2.198)	1.214 (0.627, 2.353)	0.855 (0.313, 2.333)	1.691 (0.78, 3.668)
One per week or less	1.061 (0.74, 1.522)	0.88 (0.565, 1.372)	0.433 (0.2, 0.935) *	0.799 (0.441, 1.448)
Several times per week	1.031 (0.73, 1.458)	0.773 (0.495, 1.206)	0.383 (0.166, 0.885) *	0.459 (0.248, 0.849) *
**Tobacco Consumption**				
Neither smokes nor has smoked (Ref.)	1	1	1	1
Used to smoke	0.783 (0.578, 1.061)	1.646 (1.111, 2.439) *	0.844 (0.437, 1.629)	1.327 (0.794, 2.219)
1–10 cigarettes	0.792 (0.5, 1.256)	1.178 (0.657, 2.112)	1.26 (0.549, 2.89)	0.577 (0.216, 1.54)
>10 cigarettes	1.084 (0.612, 1.922)	2.119 (1.09, 4.12) *	0.888 (0.291, 2.708)	2.407 (0.976, 5.934)
**Job Situation**				
Active (Ref.)	1	1	1	1
Retiree	0.957 (0.631, 1.452)	0.682 (0.408, 1.141)	1.114 (0.47, 2.642)	1.832 (0.807, 4.158)
Unemployed	0.923 (0.556, 1.532)	0.759 (0.418, 1.378)	0.4 (0.138, 1.16)	1.028 (0.334, 3.165)
Domestic Work	0.787 (0.488, 1.269)	0.964 (0.555, 1.674)	1.32 (0.556, 3.135)	1.93 (0.808, 4.605)
**Income**				
<600 € (Ref.)	1	1	1	1
601–900 €	0.978 (0.534, 1.79)	1.185 (0.566, 2.479)	0.343 (0.135, 0.871) *	0.901 (0.39, 2.079)
901–1200 €	0.701 (0.386, 1.271)	1.531 (0.755, 3.104)	0.374 (0.152, 0.921) *	0.759 (0.326, 1.766)
1201–1800 €	0.769 (0.423, 1.4)	0.847 (0.402, 1.782)	0.285 (0.106, 0.766) *	0.693 (0.283, 1.697)
>1800 €	0.864 (0.455, 1.641)	0.758 (0.332, 1.731)	0.158 (0.045, 0.555) **	0.756 (0.271, 2.109)
**COVID-19**				
Yes (Ref.)	1	1	1	1
No	0.701 (0.498, 0.987) *	0.778 (0.511, 1.186)	0.844 (0.428, 1.666)	0.771 (0.446, 1.331)
**Primary Attention Visit**				
No (Ref.)	1	1	1	1
Yes	1.226 (0.892, 1.684)	1.675 (1.073, 2.617) *	2.086 (0.896, 4.857)	2.005 (1.072, 3.751) *
**Emergencies Visit**				
No (Ref.)	1	1	1	1
Yes	1.03 (0.761, 1.395)	1.255 (0.874, 1.801)	1.179 (0.665, 2.089)	1.647 (1.053, 2.575) *
**Hospital Admission**				
No (Ref.)	1	1	1	1
Yes	1.138 (0.701, 1.849)	1.222 (0.696, 2.147)	1.191 (0.508, 2.793)	1.727 (0.924, 3.228)
**Specialist Visit**				
No (Ref.)	1	1	1	1
Yes	0.584 (0.442, 0.772) ***	0.814 (0.572, 1.159)	1.333 (0.748, 2.377)	1.092 (0.696, 1.712)
**Administrative Regions**				
Cadiz Bay (Ref.)	1	1	1	1
Jerez and Rural	1.199 (0.792, 1.815)	1.023 (0.598, 1.748)	0.947 (0.392, 2.288)	1.435 (0.693, 2.968)
Gibraltar Zone	0.921 (0.552, 1.535)	1.066 (0.558, 2.037)	1.082 (0.379, 3.088)	1.164 (0.471, 2.878)
Northwest Coast	0.985 (0.582, 1.664)	1.199 (0.631, 2.28)	0.889 (0.298, 2.65)	1.591 (0.667, 3.799)
La Janda	1.651 (0.96, 2.837)	1.067 (0.531, 2.144)	1.041 (0.345, 3.143)	1.223 (0.487, 3.07)
Cadiz Mountains	1.153 (0.681, 1.95)	1.22 (0.634, 2.346)	1.134 (0.395, 3.258)	1.192 (0.497, 2.859)

* *p*-value < 0.05, ** *p*-value < 0.01, *** *p*-value < 0.001.

## Data Availability

The datasets generated and/or analyzed during the current study are not publicly available due to the privacy of study participants and the health data used.

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
