# Peer review of "Multimorbidity Patterns and Their Association with Social Determinants, Mental and Physical Health during the COVID-19 Pandemic"

_ijerph, 2022, doi:10.3390/ijerph192416839_

Round 1

Reviewer 1 Report

This is an original paper that makes a significant contribution to a growing body of literature focusing on the effects of multimorbidity on physical and mental health among older adults. The analysis of the data is a strength of the paper, notwithstanding the limitations stated below.

The main limitation of the paper is that the review of literature overlooks a number of research studies of relevance. I have listed a number of these in the attached file.

A conceptual model could also be integrated (see references below).

In addition, the limitations section is brief. The limitations of a cross-sectional design needs to be acknowledged. Also, the SF-12 measure includes different dimension of health, which makes comparability to other physical and mental health measures restricted.

Here are some important references to consider:

COVID

Iaccarino, G., Grassi, G., Borghi, C., Ferri, C., Salvetti, M., & Volpe, M. (2020). Age and multimorbidity predict death among COVID-19 patients: Results of the SARS-RAS study of the Italian Society of Hypertension. Hypertension, 76(2), 366-372. doi.org/10.1161/HYPERTENSIONAHA.120.15324

Maddaloni, E., D’Onofrio, L., Alessandri, F., Mignogna, C., Leto, G., Pascarella, G., ... & Buzzetti, R. (2020). Cardiometabolic multimorbidity is associated with a worse Covid-19 prognosis than individual cardiometabolic risk factors: A multicentre retrospective study (CoViDiab II). Cardiovascular Diabetology, 19(1), 1-11. doi: 10.1186/s12933-020-01140-2

Wister, A., Li, L., Cosco, T., Best, J., & Kim, B. (2022). Multimorbidity and depression, anxiety and comprehensive impact during the COVID-19 pandemic: Analyses using the Canadian Longitudinal Study on Aging (CLSA). Clinical Gerontologist. https://doi 10.1080/07317115.2022.2094742.

Wister, A., Li, L., Cosco, T., McMillan, J., Griffith, L., & on behalf of the Canadian Longitudinal Study on Aging (CLSA) Team. (2022). Multimorbidity resilience and COVID-19 pandemic self-reported impact and worry among older adults: A study based on the Canadian Longitudinal Study on Aging (CLSA). BMC Geriatrics. 22:92. https//doi.org/10.1186/s12877-022-02769-2.

SDH & Health Associations

Marengoni, A., Von Strauss, E., Rizzuto, D., Winblad, B., & Fratiglioni, L. (2009). The impact of chronic multimorbidity and disability on functional decline and survival in elderly persons. A community‐based, longitudinal study. Journal of Internal Medicine, 265(2), 288-295. doi.org/10.1111/j.1365-2796.2008.02017.x

Northwood, M., Ploeg, J., Markle‐Reid, M., & Sherifali, D. (2018). Integrative review of the social determinants of health in older adults with multimorbidity. Journal of Advanced Nursing, 74(1), 45-60. doi.org/10.1111/jan.13408

Measurement

Cornell, J., Pugh, J., & Williams, J.W., (2007).  Multimorbidity clusters: Clustering binary data from multimorbidity clusters; clustering binary data from a large administrative medical database.  Journal of General Internal Medicine, 22(3): 419-424.

doi:10.1017/S0033291702006074.

Kirchberger, I. et al. (2012).  Patterns of multimorbidity in the aged population.  Results from the KORA-Age study.  PLoS ONE: 7, e30556.

Ng, S.K., Holden, L., & Sun, J. (2012). Identifying comorbidity patterns of health conditions via cluster analysis of pairwise concordance statistics. Statistics in Medicine, 31: 3393-3405.  doi:10.1002/sim.5426.

Author Response

Thank you very much for your revisions and suggestions. 

Please see the attachment for a detailed review of the changes made

Reviewer 2 Report

Overall, this is a very good, well written submission. The methods are appropriate, very clearly described, and applied correctly. The introduction is very good and does a fine job of describing the key concepts and problematising multimorbidity in the context of the COVID-19 pandemic.

I do however have a number of minor queries that should be addressed before publication. 

Line 95-96: "... conducted the survey through face-to-face interviews by calling telephone numbers". telephone surveys are not face-to-face.

Line 100-106: Please provide references for the facts presented in this paragraph.

Sample: More information in required on the sampling method. In particular, (1) what was used for the sampling frame (2) what was the response rate?.

Why did you chose to include only those age 50 years and why did you sensor the age groups at 69 years?

Line 166: "we need enough participants in each class". What was the minimum number needed? and was this based on fit statistics or some other criteria?

Missing values. Is it the case that all variables were missing <1% (apart from income, line 151) or did you only impute for those with missing <1%. If the latter, did you exclude cases missing data on variables with more than 1% missing or did you use FIML or similar?

Typo on Line 202 states that the highest ABIC is for the 5 class model. I assume you meant, the lowest?

Line 210: "each class was around 5%". What does the 5% refer to?

The SF-12 is a well-established measure so there was probably no need to perform the factor analysis. That said, it is done very well!

Figure 3. If it can be easily done with ggstatsplot package, the class names would be useful on the x axis. It is not a big deal so please don't waste time with this if it isn't straightforward to do.   

Line 283. A p value of zero is not possible. Please report as <0.001.

Table 3. I would be interested to see the 95% confidence intervals so as to get an idea of the precision of the estimates. The table is pretty busy as it is so you could remove the p values and simply donate significant factors in the usual form of *, ** etc. The p values themselves are pretty uninformative.

Limitations: The potential limitation of self-reported data on chronic conditions is acknowledged but an important potential implication of this should be noted. That is, the estimates for the prevalence of chronic conditions is likely to be an under-estimate.

Author Response

(The authors gave the same response as above.)

Round 2

Reviewer 1 Report

The authors have addressed all suggested revisions.